# Comparing Modern Manufacturing Tools and Their Effect on Zero-Defect Manufacturing Strategies

Peter Trebuna, Miriam Pekarcikova *  and Michal Dic

Department of Industrial and Digital Engineering, Faculty of Mechanical Engineering,
Technical University of Košice, Park Komenského 9, 042 00 Košice, Slovakia
* Correspondence: miriam.pekarcikova@tuke.sk; Tel.: +421-55-602-3244

**Abstract:** The aim of most manufacturing and production factories can be defined as achieving smart and sustainable long-term production systems, which means moving towards strategies defined by Zero-Defect Manufacturing strategies with many areas of improvement, such as: lowering overall cost and energy consumption, amount of scrapped output and wasting raw material and improving overall lead times, production status overview, and planning abilities. These facts were the motivations behind the writing of this paper. The authors considered if the classical architecture of Zero-Defect Manufacturing can be improved by additional tools conventionally used in modern manufacturing. The authors have selected Advanced Planning and Scheduling software tools. To prove that different scheduling methods can have a serious impact on overall production results, we prepared a simple case base comparing different scheduling rules. The theoretical basis for writing this manuscript was prepared by studying classical ZDM methodology and defining the industry gap. The methodology is based on the Zero-Defect Manufacturing architecture, which is essential for high-level implementation in industrial practice. Adding new tools, such as Scheduling and the Industrial Internet of Things, to the classic ZDM architecture improves overall methodology. The impact of different scheduling strategies, which is also described in this study, depends on industry and working conditions. The scheduling rules were compared by several key performance indicators, such as lead time and the number of late/unfinished orders. The study realized, with the practical accent shown this challenge, further research in connection with digitalization.

**Keywords:** manufacturing systems; scheduling; manufacturing operations; Industry 4.0; Zero-Defect Manufacturing

## 1. Introduction

Production companies have recently faced significant and important movements in the field of digital transformation and the digitalization of managing software architectures for the management and control of manufacturing production systems. This digital transformation has several different names across continents, such as Industry 4.0, Smart Manufacturing, and similar. Since the beginning of the 1990s, there has been existing software for the management of production, such as ERP (Enterprise Resource Planning) [1,2]. The capabilities of ERP systems have become insufficient in controlling, monitoring, and improving manufacturing and production management systems constantly. If this is the case, we ask if the company remains in a competitive position in its own market, depending dramatically on the company's capability to transform the core processes into more economic value. The overall economic efficiency of adding any economic value, and it follows the potential of production companies, should or rather must be found in the capabilities of the process rather than in the overall production capability. Nevertheless, the ability to deliver the required and ordered product quantity on time with the required level of product quality, keep the consumption of resources at the possible minimum level, and lowering the costs require deep operating manufacturing system knowledge. This can be

achieved by constantly collecting information and deeply understanding the behavior to apply the most effective control strategies. More than 30 years since MES (Manufacturing Execution Systems) received attention, it has been positioned as a key manufacturing software platform to be implemented in the case of "advanced control strategies", and real-time production produces control. If it comes to levels of production quality, information on several production factory levels should be available to develop integrated control solutions. The connection between the digital and physical world is becoming tighter than before, and this connection provides promising research and industrial topics, increasing the relevance of understanding the architecture that needs to be implemented to such a solution. Manufacturing Execution Systems, MES, are the centralized software (production company backbone) for the implementation of strategies of ZDM (Zero-Defect Manufacturing solutions and this goes further than traditional data acquisition solutions, such as business intelligence tools [1,3,4]. The aim is to achieve the highest possible quality of the final product in the most efficient and optimal production process.

### 1.1. Industry Gap

Across discrete industries, the biggest threat is the trend of "doing nothing"; the available research behind Industry 4.0 is extensive. The decision to invest in digital transformation and advanced Manufacturing operations tools is a strategic decision made at the main headquarters level. Such decisions influence processes across the whole organization, and industry experience shows that if the investment is not planned carefully, companies struggle to survive. The key benefit of investing in Industry 4.0 or Manufacturing Operations tools is reducing internal operating costs through the end-to-end integration of digital tools. Return of Investment is influenced by a long implementation period and high procurement costs.

### 1.2. Research Gap

Comparing modern tools, mainly software tools, with their effect on Zero-Defect manufacturing strategies opens the question of how academic research is disconnected from real industrial problems. Many companies across industries still use spreadsheet software, such as Excel, for daily short-term scheduling compared to the academia sector, which already discusses connecting discrete event simulation tools with Advance Scheduling tools. This type of company commonly collects data manually, record them into ERP solutions, and then manually triggers any change. Some of the literature, such as [2,3], already defines the scheduling in the ZDM architectures as part of Manufacturing operations. This literature defines scheduling with the following assumptions: "in case that the workorder has already started on the machine this cannot be interrupted, that mean each machine or equipment is able to handle one job at the certain defined time". In the certain moment of receiving a new order or detecting/predicting a defect, the following actions are prepared and set up as waiting to be released to the production shopfloor accordingly to a specific method; those scheduling methods take into consideration the most effective time to production rescheduling, which will include the new work task. This paper extends the research of Zero-Defect Manufacturing with the question of if conventionally used and conventionally set up scheduling methods of advanced Scheduling tools might improve ZDM architecture.

### 1.3. Research Design

By defining the Research and Industry gaps in Sections 1.1 and 1.2, the authors are connecting the classical architecture of Zero-Defect Manufacturing with Advanced Planning and Scheduling tools in a conventional setup and IIoT. The impact of the improved scheduling on ZDM is proven by a single case that compared different scheduling rules. The authors define the outcome of the manuscript as improving the architecture of Zero-Defect Manufacturing.

## 2. Materials and Methods

### 2.1. Zero-Defect Manufacturing

The product or production process problem is defined as: "in case if the output from a production is out of the range of production specification and there is no sign of indication of problem found (except the case that it is out of specifications range)—the methodology can be defined as insufficient for identifying issues, and this methodology needs to be seriously improved to be able to identify the essential challenges and problems by combining and integrating data from monitoring systems (manufacturing intelligence systems) and several important analytic methods. The classical Zero-Defect manufacturing architecture is presented and defined in Figure 1.

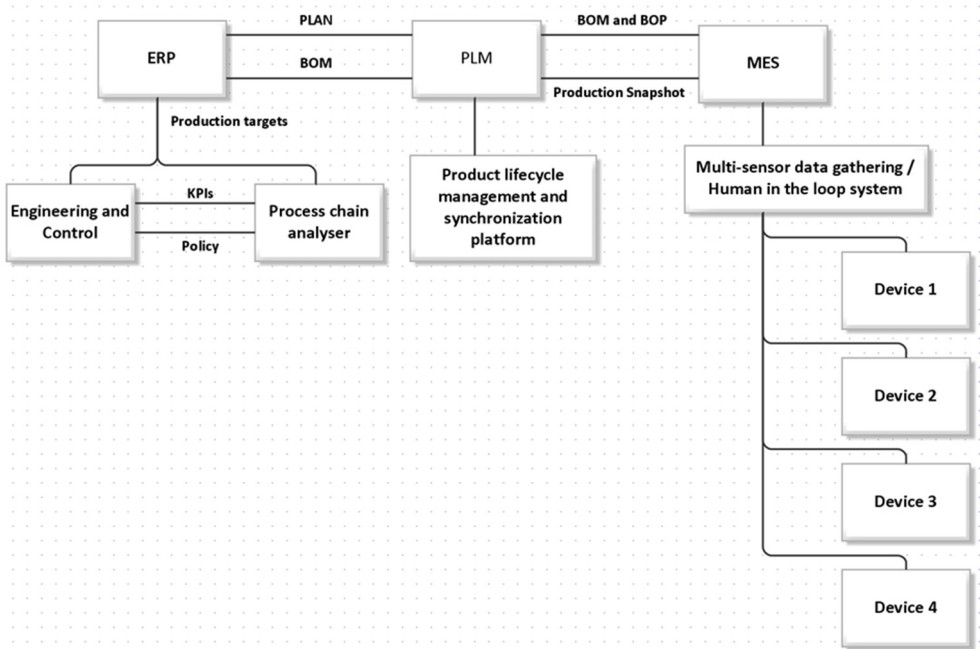

**Figure 1.** Zero-Defect manufacturing classical architecture.

It might be defined as the combination and integration of the following statements into strategies of the Zero-Defect Manufacturing (ZDM) approach [1,2,5]:

- Continuous process of quality control of production;
- Collaborative manufacturing;
- Monitoring of the key process parameters;
- Sharing of manufacturing data without media breaks along the whole supply chain;
- Measuring of an input to the production process;
- Measuring of output with automatic sorting;
- Predictive maintenance (online);
- Data management and analytics;
- Re-configuration and re-organization of production setup.

Zero-defect manufacturing strategies shall be defined as a viable concept of sustainable production, but the research in this field already considers Zero-Defect Manufacturing from a different angle of perception. The industrial and academic sectors need to standardize the terminology in the Zero-Defect Manufacturing field is recognized by industry or academic or research experts. The actual study tries to provide a clear definition of Zero-Defect Manufacturing to be used across the scientific or industrial communities, thus eliminating any misunderstandings that exist [5].

*"ZDM is a holistic approach for ensuring both process and product quality by reducing defects through corrective, preventive, and predictive techniques, using mainly data-driven technologies and guaranteeing that no defective products leave the production site and reach the customer, aiming at higher manufacturing sustainability".* [5]

The classical Zero-Defect Manufacturing Framework is defined in Figure 2. In the beginning, two pairs in the diagram, defined as Detect–Repair and Detect–Prevent, are not totally modern. They have existed in the manufacturing domain for decades already. However, what is new, are the modern technologies that are more advanced and data-driven, enabling more efficient implementation. Many academic publications [for example, "Sousa, J.; Mendonça, J.P.; and Kiritsis, D. Zero-defect manufacturing the approach for higher manufacturing sustainability in the era of Industry 4.0: a position paper" mention in Section 2] refer to detect–repair as a corrective approach, which defines something happening as already faulty and requiring corrections. Nevertheless, pair: "Detect–Prevent" might be referred to in any publications (academic, industrial) as an approach with a preventive character, and it uses production data and information to avoid possible future defects. Nevertheless, "predict–prevent" is a new concept that uses production data and is an advanced data-driven pair. In almost every definition of Zero-Defect Manufacturing strategies, a "fault" is defined as a defect, or it can also be called the non-fulfilment of a requirement related to an intended or specified use. This concept predicts when a fault (defect) will occur in the future, enabling it to be more ahead and prevent a fault before it occurs. The strategy follows the way of thinking of 'doing things right the first time,' and the remaining two follow triggering an action after the fault (defect) appears. Strategies of Zero-Defect Manufacturing follow two similar production approaches: one is product-oriented, and the second is process-oriented. Both lead to the same outcome: minimizing the defects and eliminating defects to zero at the end of the production line [1,2,5].

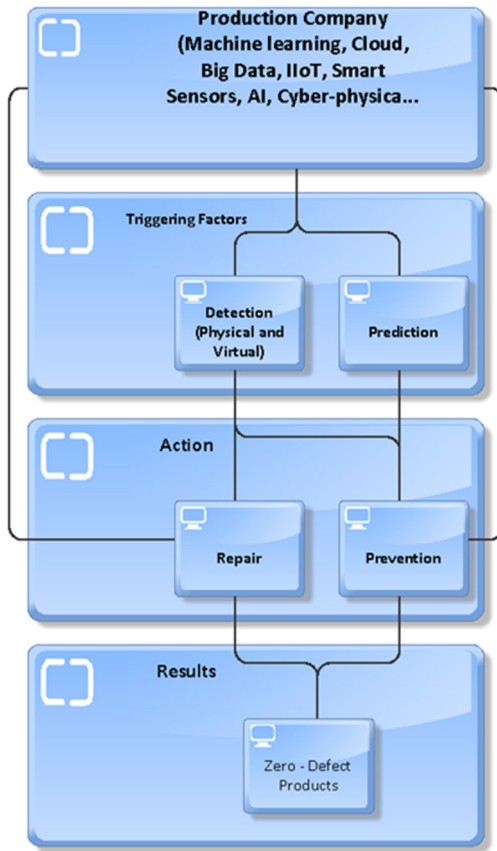

**Figure 2.** Classical Zero-Defect Manufacturing Framework.

### 2.2. Challenges to Overcome While Implementing Zero-Defect Manufacturing Strategies

As Zero-Defect Manufacturing is a strategic framework, the decision-making process is highly relevant. Decision-making choice while implementing process innovations is possible [6]. Thereby to the level of ambiguity and analysability of decision-making. The actual view of decision structuredness claim that there are no higher decision-making methods [6]. However, a decision-making method might be more fitting to specific conditions, and this leads to an effective result under these conditions [7]. In some research papers [8], the authors mention the wide-spectrum views of the authors that are important in academia or industry, i.e., Zero-Defect Manufacturing strategies compared with traditional Quality improvement methods and provide real facts that are important for navigating manufacturers and producers to adopting Zero-Defect Manufacturing strategies, and this part of the study tries to provide several backed answers to commonly asked questions that might be open about Zero-Defect Manufacturing approaches [1,2,5]:

- Is it possible to save "money" while implementing ZDM? (Section Improvement of KPIs while implementing ZDM Strategies);
- What is the role of Manufacturing Operations management in ZDM strategies? (Section: Manufacturing Operations management/Manufacturing Executions system in the environment of ZDM (Zero-Defect Manufacturing));
- Role of Scheduling tools in ZDM (Section: Scheduling);
- Is it possible to balance machine utilization? (Section: Scheduling);
- Can scheduling optimization improve the overall lead time of production? (Section: Scheduling);
- Is it possible to process large amounts of data in the company? (Section Industrial Internet of Things and its impact on Zero-Defect Manufacturing);
- Is ZDM improving the traceability of the supply chain? (Section Improving traceability by Zero-Defect Manufacturing).

### 2.3. Improvement of KPIs While Implementing ZDM Strategies

Traditional Quality Improvements methodologies (such as the LM, the L6S, the SS, the TOC, and the TQM) were already introduced many years before and have served as the paradigm of mass production [9–11]. At this moment, the production environment is changing significantly and is shifting more and more from mass production to product-oriented customization and, for some specific scenarios, is moving towards personalization [12–14]. The market is pushing to reduce product life cycles to be able to capture the needs coming from the market. There is a constant need to cut the time required for market development. Additionally, batch sizes have been significantly reduced, in the era of mass customization and personalization, a consequence of which is the increased defects during production as a result of less time being available for optimizing production and implementing Traditional Quality Improvements (QI) [15–18].

Traditional QI methods were designed using the 'corrective' paradigm as a driver, meaning that in order to be applied, there must first be a problem in order to solve it. ZDM comes to fill this gap using the 'prediction' strategy, which identifies the problem before it happens, providing the time for reacting and preventing the problem from being created. Furthermore, usually 100% inspection, no defective products will leave the factory; thus, the negative environmental impact of production is reduced while not increasing the production burden [1,2].

### 2.4. Manufacturing Operations Management (MoM)/Manufacturing Execution System (MES) in the Environment of Zero-Defect Manufacturing

The Manufacturing Execution System (MES) is part of the vertical production management software, which is programmed and designed to meet the production needs of a factory by vertically connecting the administration part of the company with production management and systems to control the production and products. The role of MES is also closely connected to the outputs of the three levels (layers) of the information sys-

tem, especially those that require functions in planning, such as Manufacturing Resource Planning, Executive functions or quality control systems, and systems for supervisory control, and the functions of data visualization, so the management of production has a full view and visibility to the certain information that exists within the organization. [19–22]. Those systems have important roles in implementing Zero-Defect Manufacturing strategies. MES′s key function is order management, but from a broader perspective, we can define more functionalities, such as the allocation of resources, long-term and short-term planning (Scheduling), the management of processes, quality control management and several operation analyses, and visualizations. All of those functions operate to translate the data generated in real-time by production into useful information from the process and management standpoint. This is being further introduced into other adjunct processes that can be understood as materials and equipment management, the traceability of products, and systems that support documentation and quality (Figures 3 and 4) [23–26].

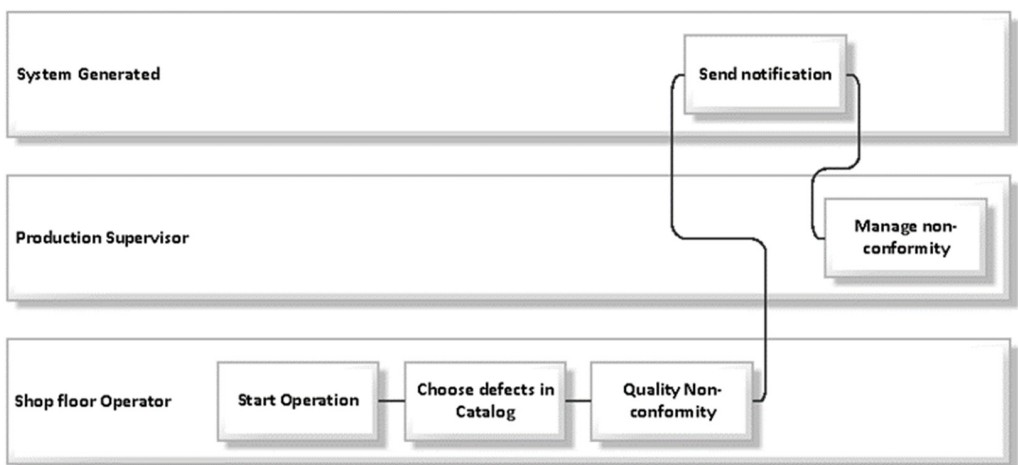

**Figure 3.** Defect declaration scheme [27].

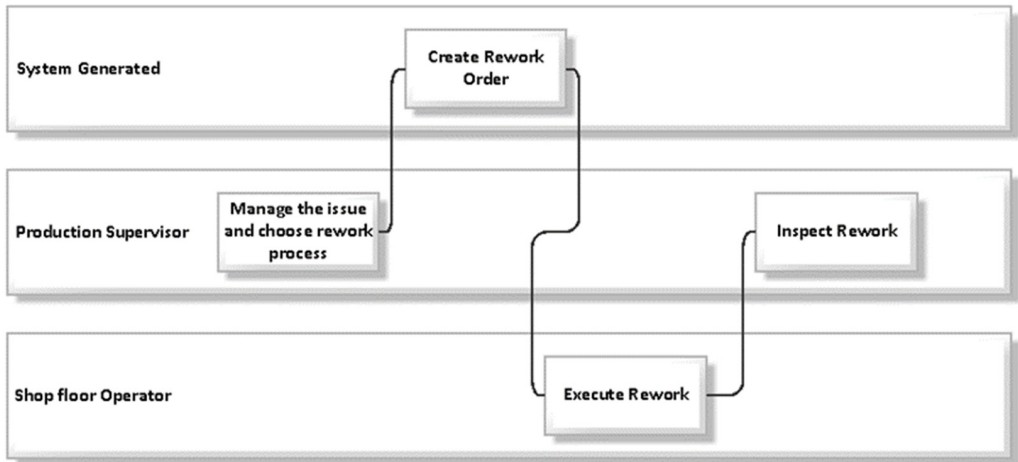

**Figure 4.** Rework triggered by production supervisor scheme [27].

The MES systems help the operator to communicate with his production supervisor any defect directly through the system (Figure 5). Most of the commercially released MES allow the pause of operation, declare a defect directly from the operator's place, and continue with other jobs until the production supervisor decides what to do with the rework.

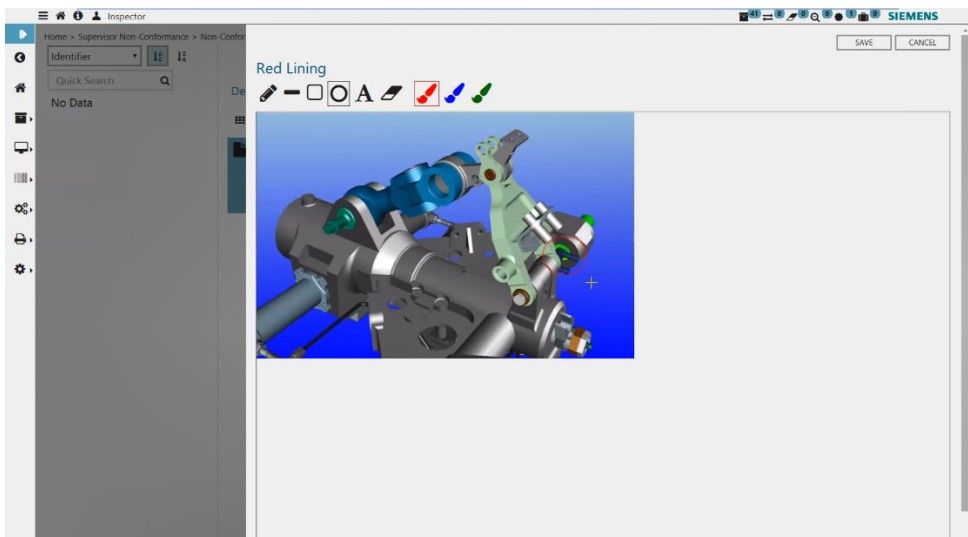

**Figure 5.** MES view-defect declaration-red circle is operator´s note.

The position of any MES is to support and create faultless processes for production and to support the creation of a consistent view of the data generated by production. We can define the other benefits of MES/MOM systems in Zero-Defect Manufacturing strategies as follows:

- Production traceability;
- Downtime reduction, nonconforming production;
- Shortening of setup times;
- Increasing OEE (Overall Equipment Efficiency);
- Inventory reduction;
- Paperless production;
- Ensuring the accuracy of production data.

### 2.5. Advance Scheduling Tools-Literature Review

APS, also known as Advanced Planning and Scheduling systems, are defined as manufacturing management systems alongside the ERP system that optimally allocates production capacity to meet customer (internal or external) demand. If simpler planning methods are not able to fulfill the more complex problems between priorities and where response time against production deviation is crucial or where the complexity is too high, then APS Systems are well suited. APS is supporting companies towards more accurate and precise accurate scheduling, eliminating the creation of not needed parts in time and only delivering parts at a time when they are needed to support the Zero-Defect Manufacturing methodology. Short-term planning/scheduling is especially very challenging due to the high number of different schedules that are possible with even a few numbers of items to be produced, and it can be understood as one of the most important processes for every production factory, but it is very different from company to company [23–25,28]. All scheduling processes share several common features. The process of accurate scheduling is triggered by the scheduling supervisor-the person who prepares the schedule on a regular basis. He creates a request for production-then the system generates a new work order that is combined with the current production snapshot. Once the schedule is prepared-this is then forwarded to production as work orders for execution. Once there is an identified deviation plan-reality-the rescheduling process is triggered. The production supervisor needs to verify this deviation and then request the production scheduler to reschedule the work orders. The advanced planning and scheduling machine utilization view and the process of work order scheduling are shown in Figure 6 [23–26,29].

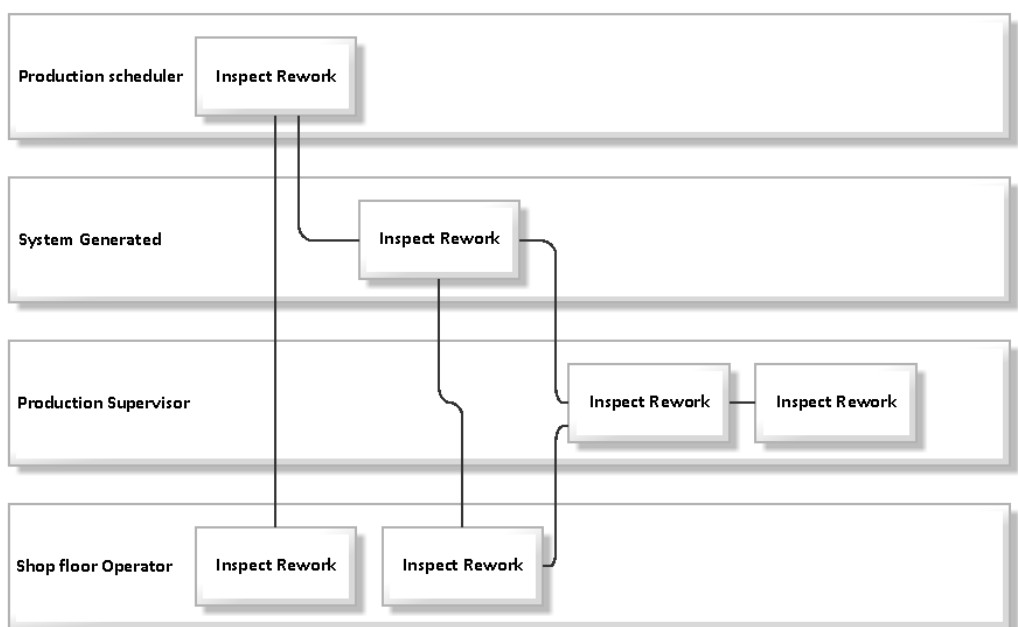

**Figure 6.** Process of work order scheduling scheme [27].

2.5.1. Advanced Scheduling Tools-Scheduling Optimization-Single Case to Compare Different Scheduling Rules and Its Impact on Production Results

Schedule optimization can be defined as the process of achieving every individual task or process in short-term planning that is in line with the overall company target. Schedule optimization can be used by any organization or business alike to define the highest priorities at the leading place while setting times for tasks to take place [29,30].

The priorities of schedule optimization can be set individually by organizational needs. The schedule can optimize the average lead time, machine utilization, or the ratio of finished work orders to incomplete/late work orders. To achieve Zero-Defect Manufacturing, delivering parts on time and optimizing the setup of the machines is an important Key performance indicator. Using Siemens Opcenter Advance Planning and Scheduling, companies can compare the scheduling scenarios that have the best ratio.

**Forward scheduling** can be defined as the short-term scheduling of production, which is moving forward from a certain starting point (date). The key aim of this process is to finish each work order on the schedule as soon as the resources that are necessary for completion are available, with minimum or even no waiting times between two separate tasks. In cases where resources are already available, the task is considered complete, but in cases where the resources are not available yet, the work order (task) is put on hold status-and this means that the project is on pause-until the moment when resources are ready. Forward scheduling does not allow for the generation of incomplete work orders, but it puts them on late status, as visibly reported in Figure 7 [24].

**Backward scheduling** is when production produces their items at the last possible available period before the due date. The order starts with a planned receipt date or due date-one that is usually defined upon the customer's (internal or external) order (see Figure 8) [24].

**Grouping work orders by the material** that enters the manufacturing process can be a means to optimize production and its overall results. Grouping work orders leads to minimizing set-up times, which has positive effects on the overall scrap production-which is also a step-forward Zero-Defect Manufacturing strategy. Material class grouping is a visible change in the Gantt charts of scheduling tools. In comparison to forward scheduling, scheduling with material class grouping is more effective in terms of late orders and average set-up times, as shown in Figure 9 [23–25].

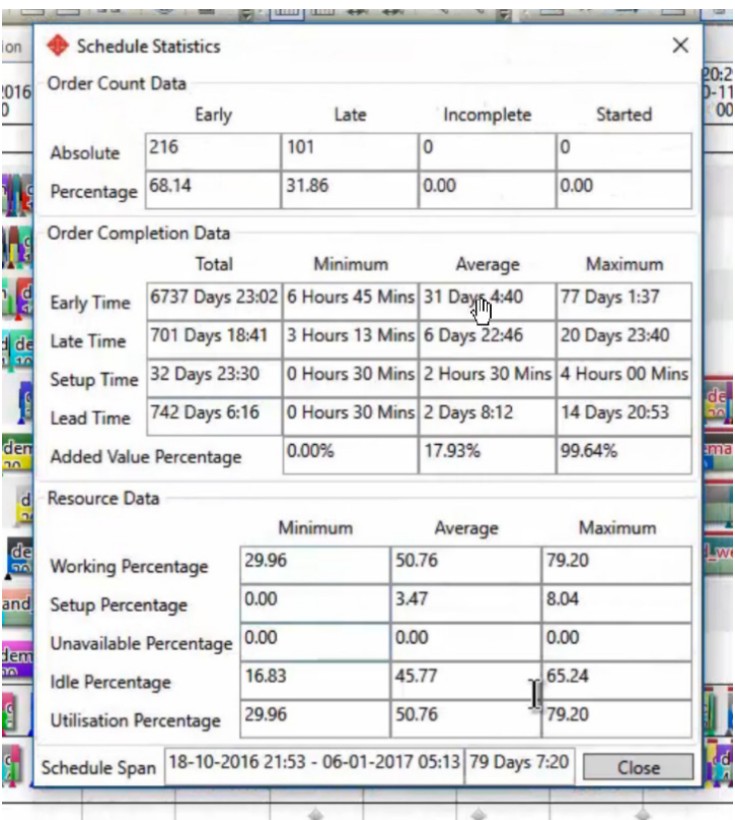

**Figure 7.** Forward Scheduling-Siemens Opcenter APS preview.

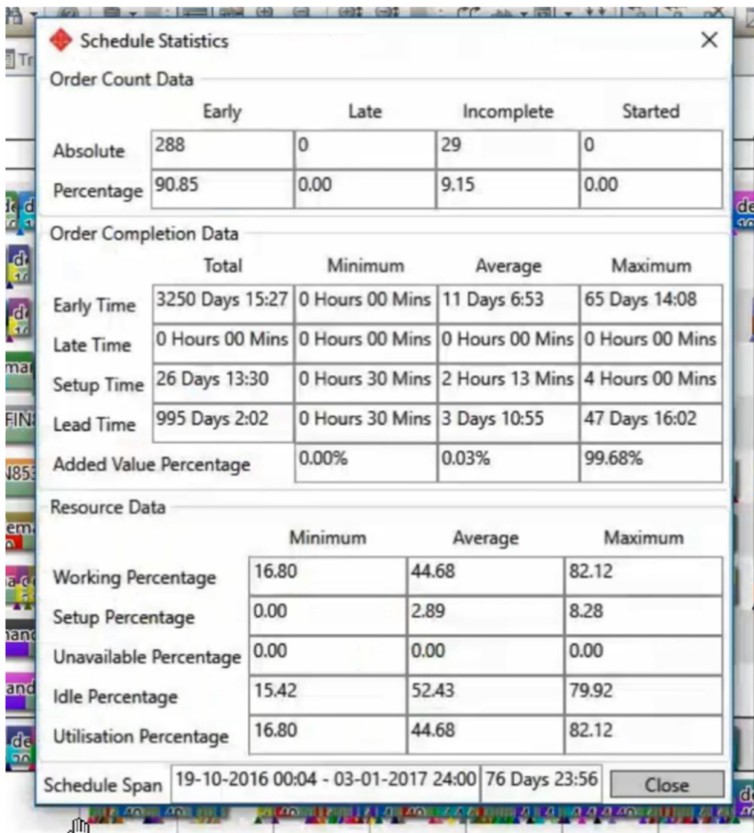

**Figure 8.** Backward scheduling-Siemens Opcenter APS.

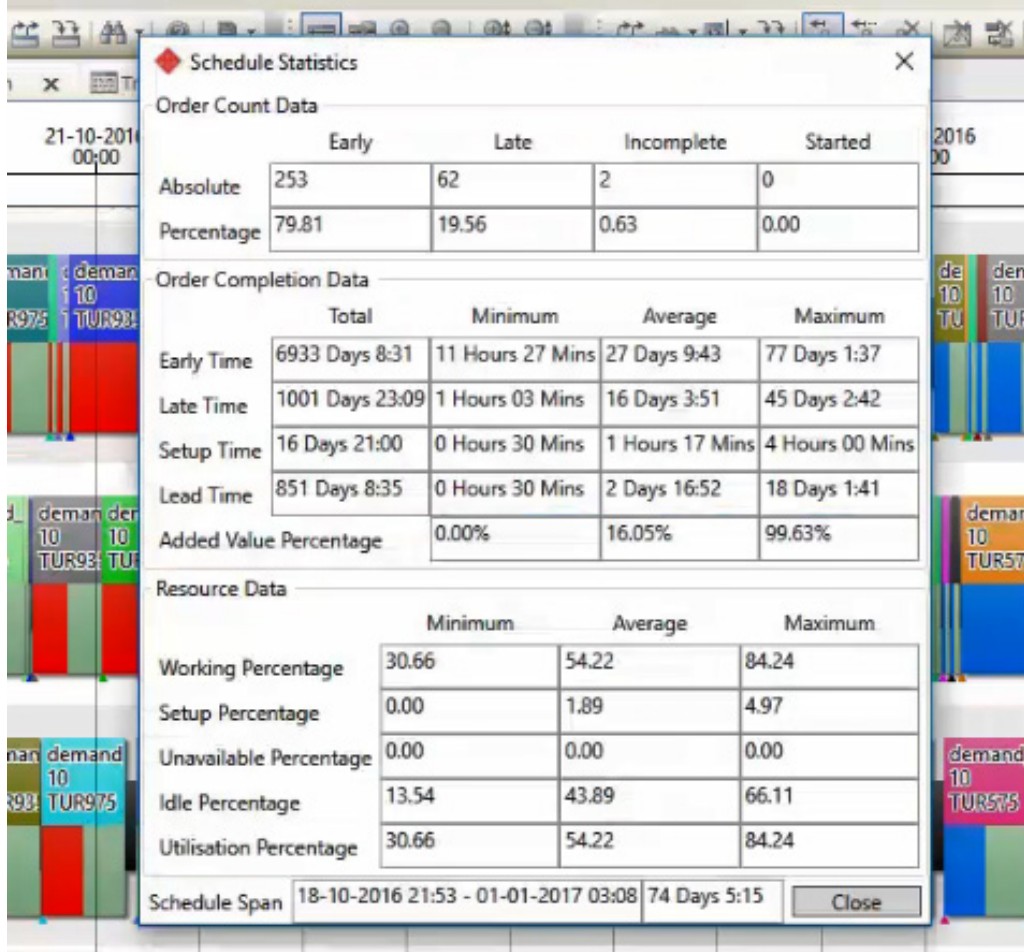

**Figure 9.** Material glass grouping-results-Siemens Opcenter APS preview.

Mixed scheduling strategies are also known as **forward scheduling with preferred sequences** (Figure 10). This strategy maintains the APS rules that are the preferred sequence for late orders, as we see that late orders are usually in specific time ranges, and the rest is scheduled with a forward rule. This mixed strategy allows for the elimination of late orders => increasing the throughput of the production.

Comparing all four optimization rules of scheduling, it is visible that using combined optimization rules of preferred sequences to eliminate late orders with the advantage of forward scheduling is the option to optimize the production schedule, which is more efficient with the schedule leads elimination of the setup times and creating scraps. As Table 1, Figure 11 demonstrates-eliminating late orders or incomplete orders by increasing the average lead time is not the most effective means of production.

**Table 1.** Comparison of scheduling operations and its results.

| Scheduling Rules | Late (Products/Parts) | Incomplete (Product/Parts) | Avg Lead Time (Hours) |
|---|---|---|---|
| Forward scheduling | 101 | 0 | 54 |
| Backward scheduling | 0 | 29 | 83 |
| Material Class grouping optimization | 62 | 2 | 65 |
| Preferred sequence (resource based) forward scheduling | 29 | 0 | 50 |

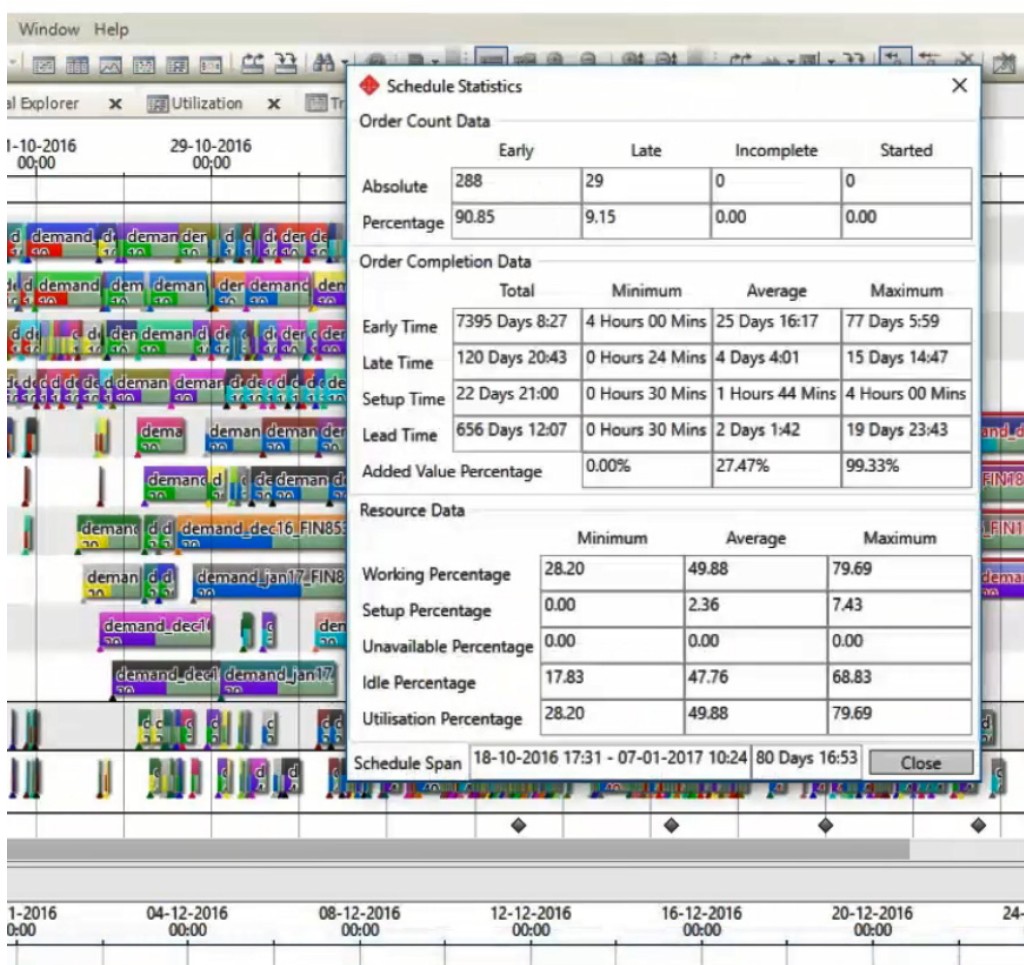

**Figure 10.** Forward scheduling with preferred sequence-Siemens Opcenter APS.

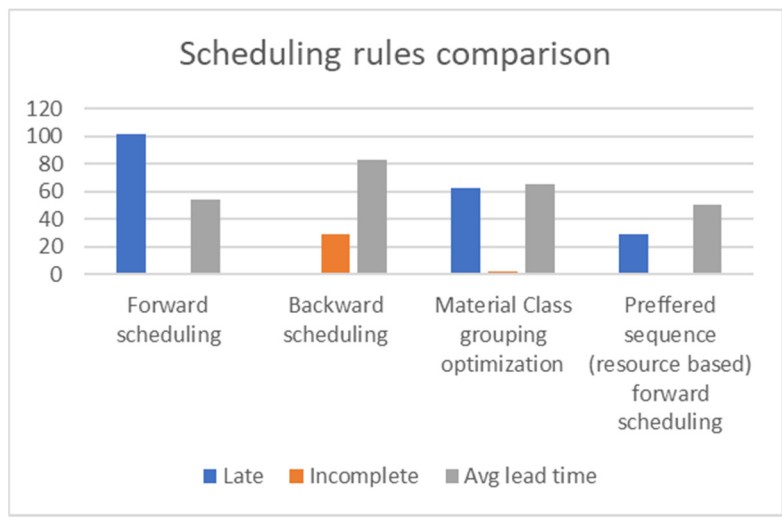

**Figure 11.** Comparison of scheduling operation-graphical.

2.5.2. Input Data to Compare Scheduling Optimization Strategies

The data used for this single case are example data. A single-use case can be widely adopted by many discrete production companies, but scheduling rules and models shall be adopted.

Production Type: Discrete manufacturing, the automotive sector as Tier 1 supplier for OEM. To define the input data, the definition of challenges that are more common for this type of company.

- Agile Delivery:

    Prioritize agility to quickly respond to demand changes;
    Focus on capacity management and the efficient utilization of resources.

- Supply chain complexity:

    Efficiently manage supply chain;
    Manage production output on a day-to-day basis.

- Product complexity:

    Increasing number of product variations and configurations;
    Produce globally, sell locally;
    Market differentiation by country and vehicle segment.

To compare the scheduling rules, the input data for the model needs key attributes and parameters, such as process times, constraints, and scheduling rules. Preview of resources-machines in Siemens Opcenter APS presents in Figure 12.

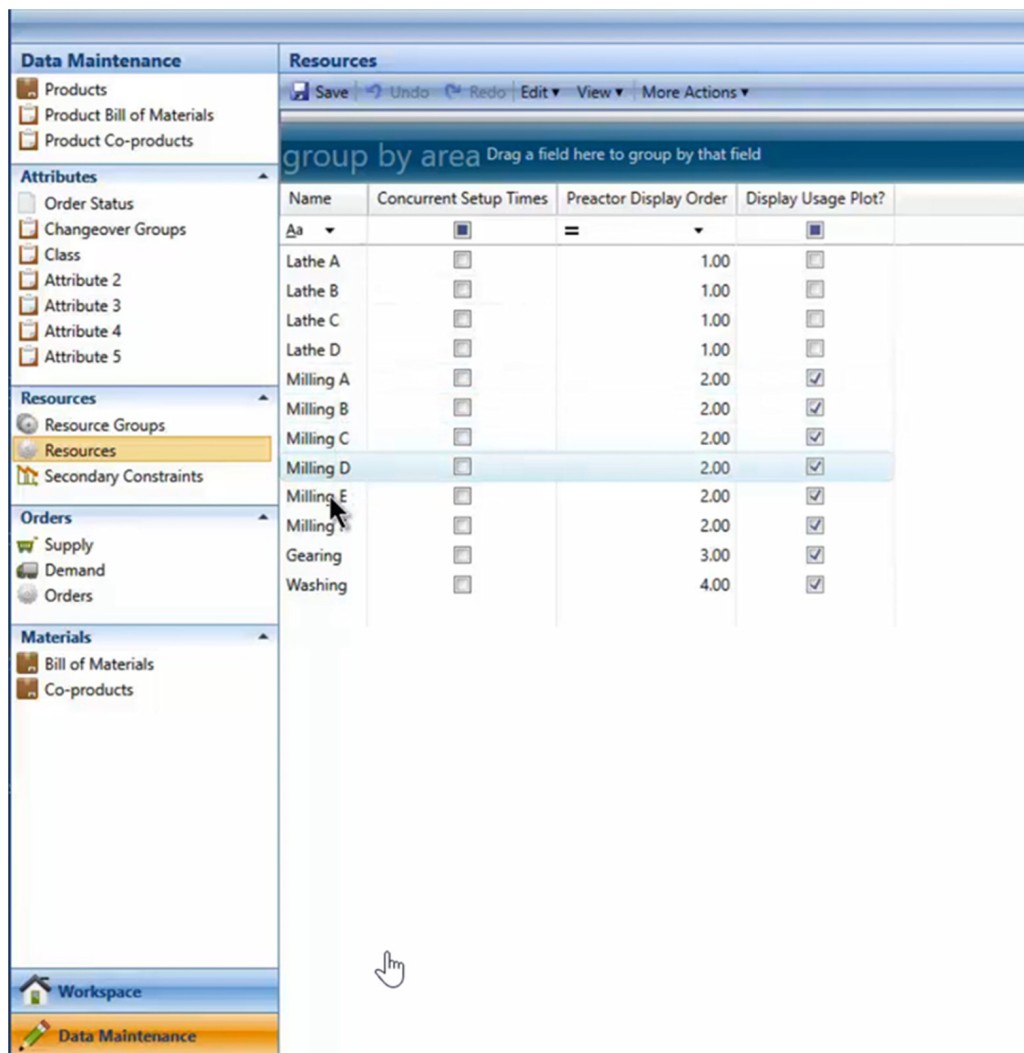

**Figure 12.** Preview of resources in Siemens Opcenter APS.

- Resources, processes, and product configurations:
  - Product;
  - Orders;
  - Resources.
- Constraint modeling:
  - Setup time;
  - Production constraints;
  - Calendars.
- Scheduling:
  - Sequencer;
  - Forward scheduling;
  - Backward scheduling.
- Optimization:
  - Scheduling rules;
  - Heuristic scheduling rules.

Once the challenges and raw input data are defined, constraint modeling is important to achieve an accurate schedule. Table 2 shows the constraints that were taken into consideration while defining the model to compare the strategies. Once we defined the constraint model, we needed to define the work order flow throughout production, as shown in Figure 13.

**Table 2.** Constrains model to create model for scheduling.

| Supply (RAW Materials) | Turning | Milling | Gearing | Washing | Finished Goods |
|---|---|---|---|---|---|
| | Changeover mgt | Changeover mgt | Changeover mgt | Changeover mgt | |
| | Dedicated resources | Dedicated resources | Dedicated resources | Dedicated resources | |
| | Operators | Operators | Operators | Dedicated resources product type preference | |
| | | Tool constraints | | Operators | |

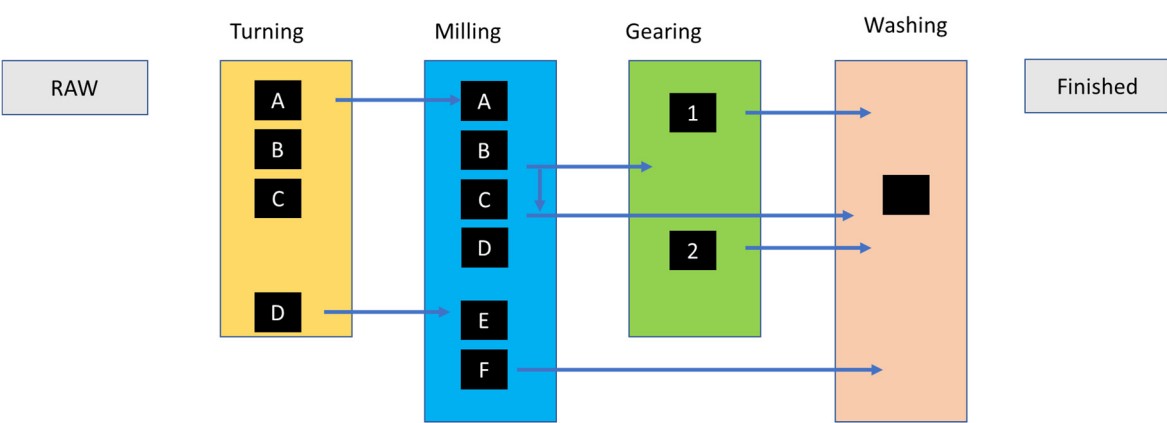

**Figure 13.** Process of work order flow.

Figure 14, Similar types of machines can use similar types of operations but with some differences. These machines can be replaced in some cases for better machine utilization. If the machine is using specific unique tooling-this tooling can be considered a secondary constraint, as scheduling tools differentiate machines as constraints with different parameters (see the constraint model in Table 2).

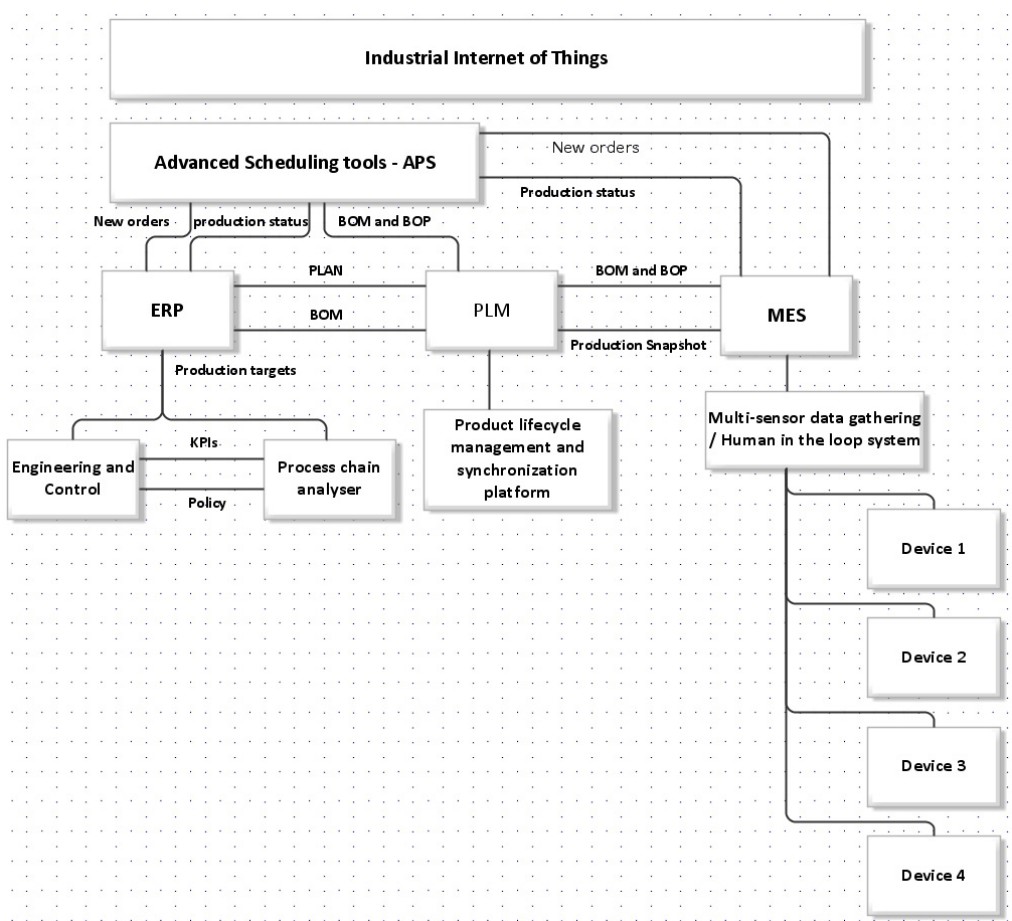

**Figure 14.** Zero-Defect Manufacturing classical architecture extended by IIoT and Advance scheduling.

### 2.6. Industrial Internet of Things (IIoT) and Its Key Importance on Zero-Defect Manufacturing Strategies

We might position IIoT as even more important in enabling access to equipment and different work machines, which were in the past hidden in separate manufacturing silos. Such a connection might be called IIoT and enables better control and more predictive operation and maintenance of the machines and tooling. We can define it as the evolution that allows IT (Information Technology) to access even higher levels of digitalization in manufacturing systems, and it is extending classical pillars of the Zero-Defect Manufacturing strategies landscape. The Internet of Things enables manufacturing companies to access a new range of applications that can run around the shop floor. The range is from the connection of the shop floor to the smart grid or sharing the production facility as a service, or it is also enabling much more flexibility and agility within entire production systems themselves [3,29–34]. It can be defined as an evolutionary step towards "Smart Factories", which require access to any external parties that interact with manufacturing systems enabled by IoT. This is a form of connected industrial systems that communicate and coordinate their analytics and actions to improve production performance and production efficiency as well as reduce or eliminate downtimes. Services connected with the manufacturing world do not need to be defined in an intertwined or linked manner to any physical system, but they are rather run as services in a shared physical world. We might define some of the most important challenges related to implementing any cyber–physical systems, which include network integrations, affordability, or the interoperability of any engineering system. The majority of production factories are struggling to justify investments that might be risky or expensive or uncertain investments to systems of smart manufacturing across the company. The change in the organizational structure or culture of manufacturing is

occurring slowly, which also hinders technology integration [17,35–38]. Nowadays, factories contain pre-digital age control systems, which are not frequently replaced due to the fact that they are still serviceable. Adding cyber–physical systems to those pre-digitalized control systems as retrofits is difficult and expensive. The IIoT is commonly selected as the key protocol to make heterogeneous distributed systems that efficiently interact with the usage of even-driven frameworks. The dependency of collaborative systems is based heavily on data sharing, but the importance is that the most important parts are autonomous or semi-autonomous, which are data-driven (data-driven decision-making) [1,2,25,26].

## 3. Results and Discussion

Comparing outputs and their impacts on overall Zero-Defect Manufacturing strategies, it is visible that the classical architecture of ZDM is not sufficient in modern manufacturing, influenced by many external factors such as pressure on cost, human resources challenges, the extreme cost of energy, and many more. All of those factors create environments that push manufacturing companies to extend the three main pillars with new tools. Figure 14 shows the improved Classical architecture of Zero-Defect Manufacturing by implementing Advance planning and scheduling tools and the Industrial Internet of things. Advance Planning and Scheduling tools communicate bi-directionally with every pillar. The proposed architecture of ZDM also shows the almost ideal digital roadmap for most of the companies in the category of SMB-Small and Medium Businesses. It takes new orders and unfinished status from the ERP system, BOM and BOP are communicated from the PLM layer, and the production status and communication with the shop floor are achieved by the communication between the Advance Planning and scheduling tools and MES. The scope of the Industrial Internet of Things is almost unlimited, and its implementation depends on the industry and its needs. The capabilities of the Internet of Things to collect data and use tools to analyze data predictively can predict machine/tool breakdown and eliminate creating scraps or damages caused by worn tooling. The right implementation of the tools of APS and IIoT will improve the overall quality of the output as well as product traceability. The consequences of improved traceability are a faster reaction to non-conformity products and avoiding situations with low-quality shipping.

## 4. Conclusions

This paper has introduced, analyzed, and compared modern tools and methods that might be used further for Zero-Defect Manufacturing strategies and then lead to improving the overall digital strategy of production companies. Improving the classical architecture of ZDM (see Figure 1) with Advance Planning and Scheduling (APS) tools and Industrial Internet of Things (IIoT) (see Figure 14) is improving not only the overall methodology but also creating a digitalization roadmap for production companies. Benefits and acceptable return of investment (ROI) after implementing Zero-Defect Manufacturing strategies alongside the tools described in this paper will bring benefits mainly in the production companies world, which is in charge of high-value-added parts (more sophisticated parts or parts from more expensive materials and similar). The discussed single case comparison between the different scheduling rules (forward, backward . . . ) on our data showed how different production types are sensitive to different methodologies. This fact leads the authors to the conclusion that every discrete production is different, and it is almost impossible to generalize recommendations. The main difference between different scheduling rules is the relation to incomplete/late orders. We might define the key challenge as the period of the ramp-up in which the actions are starting to be implemented into the company processes. Some of the literature [1,2,39] has already mentioned implementing APS into the operation layer of the ZDM architecture; the position of the authors is that every APS algorithm that improves overall efficiency, lead time, and the number of set up is already an improvement of the classical architecture of ZDM (see Figure 14). Companies belonging to the category of SMB (small and medium businesses) are starting their digitalization journey with "low-hanging fruit", so digitalization parts with the best Return

on Investment. The improved classical architecture of ZDM with tools of APS and IIoT might be an interesting roadmap for SMB discrete manufacturing companies. The major challenge for future production companies or companies in the future will be the rapidity and the right sequence of actions with which those companies can reach a steady use and fast adaptation of the Zero-Defect Manufacturing strategies. One of the conclusions is also that presented software tools are just the initial step of consideration while improving the quality of production outputs and are part of the bigger digitalization movement.

**Author Contributions:** Conceptualization, M.D.; methodology, P.T. and M.P.; software, M.D.; validation, M.D., P.T. and M.P.; formal analysis, M.D. and M.P.; investigation, M.D., P.T. and M.P.; resources, M.D.; data curation, M.D.; writing—original draft preparation, M.D.; writing—review and editing, P.T. and M.P.; project administration, P.T. and M.P. All authors have read and agreed to the published version of the manuscript.

**Funding:** This research received no external funding.

**Institutional Review Board Statement:** Not applicable.

**Informed Consent Statement:** Not applicable.

**Acknowledgments:** This article was created by the implementation of the grant project APVV-17-0258 Digital engineering elements application in innovation and optimization of production flows, APVV-19-0418 Intelligent solutions to enhance business innovation capability in the process of transforming them into smart businesses, VEGA 1/0438/20 Interaction of digital technologies to support software and hardware communication of the advanced production system platform, KEGA 001TU-KE-4/2020 Modernizing Industrial Engineering education to Develop Existing Training Program Skills in a Specialized Laboratory. VEGA 1/0508/22 Innovative and digital technologies in manufacturing and logistics processes and systems.

**Conflicts of Interest:** The authors declare no conflict of interest.

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
