# Peer review of "Comparing Modern Manufacturing Tools and Their Effect on Zero-Defect Manufacturing Strategies"

_applsci, doi:10.3390/app122211487_

Round 1
Reviewer 1 Report (Previous Reviewer 3)
1. Concise the abstract.
2. The novelty (if/any) needs to be appeared in the paper.
3. Concise the conclusion with recent references.
Author Response
Dear Reviewer,
please find attached the answer to your review,
best regards
Authors

Reviewer 2 Report (Previous Reviewer 2)
The submitted manuscript “Comparing modern manufacturing tools and their effect on Zero-defect Manufacturing strategies” corresponds to the special issue’s scope. The paper presents a comparative analysis of modern tools and methods for Zero-defect manufacturing strategies. As a result, it leads to the improvement of the overall digital strategy for production companies.
However, in my opinion, the manuscript has several drawbacks and needs minor revision:
1. It will be better to use the traditional structure for MDPI-Applied Sciences papers (Introduction – Materials and Methods – Results and Discussion - Conclusions). It will help significantly in presenting the scientific novelty of the research.
2. Please explain in the text the meaning of “A”, “B”, … “F” (Fig. 14, 15), and “1” and “2” (Fig. 15). If it is the workpieces that need machining on the different operations, please provide at least the brief information (geometric form, accuracy, roughness, machining time, etc.).
3. The figures are unreadable and must be improved (Fig. 6, 7, 8, 9, 10).
4. Fig. 3 and Fig. 4 need references.
5. Table 1 on page 12 must be appropriately formatted.
Author Response
Dear Reviewer,
please find attached the answer to your review,
best regards
Authors

Reviewer 3 Report (Previous Reviewer 1)
The authors have clarified all questions raised by the reviewers. After the revision the quality of the manuscript is good. The readability of the manuscript is improved.
So article may be acceptable in its present form.
Author Response
Dear Reviewer,
please find attached the answer to your review,
best regards
Authors

Reviewer 4 Report (New Reviewer)
The paper is somewhat theoretical, chapter 2 being very extensive. I think that more contributions by the authors should appear!!!
There are also some mistakes such as:
1) too much empty space between lines 299-300, 310-311, 317-318.
2) table 1 is not visible in its entirety. I think it needs to be redone.
Then some mistakes appear in the text:
1) line 51 instead of in order to, must be written to;
2) line 399 must have a comma after the constraints:
3) line 419 instead of is connected with, must relate to;
4) line 435 instead of and also, it must be and.
Author Response
Dear Reviewer,
please find attached the answer to your review,
best regards
Authors

This manuscript is a resubmission of an earlier submission. The following is a list of the peer review reports and author responses from that submission.
Round 1
Reviewer 1 Report
Journal Name: Applied Sciences
Manuscript ID: applsci-1804385-peer-review-v1
Manuscript Title: Compering modern manufacturing tools and their effect on Zero-defect Manufacturing strategies
Comments to Authors
This article deals with presented and studied current technologies and processes that may be applied further for Zero-defect manufacturing strategies, leading to an improvement in the entire digital strategy of manufacturing businesses As major architecture modules, software tools such as an ERP and a MES were provided in terms of software modules and integration with other management systems. Benefits and an acceptable return on investment from applying Zero-defect Manufacturing methods in conjunction with the technologies outlined in this article will benefit mostly manufacturing organisations in the globe that are in charge of high value-added items are investigated, which is an interesting topic.
This work is well-written and of high scientific merit. The suitability of the research procedures was employed, as well as a thorough understanding of research methodology. The research work was properly planned and carried out. For the kind of literature discussed in this article, in-depth knowledge of the area is required. The paper's results and interpretation should be enhanced.
After reading the manuscript, I can conclude that this paper may be accepted with a Minor revision.
However, few modifications are needed to further enhance the quality of the manuscript.
- Abstract of the article is not clear and concise. The abstract part needs to include mathematical findings to be more informative.
- Title of the manuscript to be corrected in revision.
- In the introduction section add recent literature published after 2018.
- Results and discussion must be supported by standard literature.
- Figures 5, 6, 8-13 and images are poor. It is hard to see and investigate. High-quality Figures to be provided for better readability with proper legend and labels.
- The conclusion is needed to write more precisely with the application of these existing methodology.
Author Response
Dear Reviewer 1,
please find attatched answer
Best regards
Authors

Reviewer 2 Report
The submitted manuscript corresponds to the scope of the Special Issue “Manufacturing Systems Operations and Engineering”. The paper deals with the urgent problem of zero-defect production, which is topical and timely regarding sustainable development goals and current industrial challenges.
However, in my opinion, the paper has several drawbacks and needs minor revision.
1. The paper title has a mistake. It should be “Comparing” instead of “Compering”. Alternatively, it can be changed to “Comparative Analysis of the Modern Manufacturing Tools and Their Effect on Zero-Defect Manufacturing Strategies”.
2. The keywords should be rewritten. Avoid duplication of phrases/terms used in the title. Add at least one keyword from the SDG list.
3. The paper looks like a book chapter. It should be structured according to one of the well-known approaches (e.g., Introduction, Literature review, Research Methodology, Results, Discussion, Conclusions.
4. The research gap must be clearly specified, and the research aim should be stated in the Introduction.
5. The abbreviation “QI” must be specified in the text before line 156.
6. It is difficult analyzing the results presented in Fig. 8-12 without input data.
7. It is desirable to compare the obtained results with other scientists’ data. Appropriate citations must be added in this chapter. Moreover, please describe the solved research gap.
8. Conclusions should be improved. Please highlight the scientific novelty of your research, describe the practical value, please accent quantitive and qualitative data. Further research tasks can be announced.
Author Response
Dear Reviewer 2,
please find attached answer
Best regards
Authors

Reviewer 3 Report
The paper has good work done for compering modern manufacturing tools and their effect on Zero-defect Manufacturing strategies but there are some items should be followed to be available for publishing in MDPI, applied sciences journal as follows:
1. Abstract is weak and needs to be reformulated and concise.
2. The novelty (if/any) needs to be appeared in the paper.
3. ADD more details for explanation traditional techniques and the NEW proposed technique.
4. The paper needs more samples to discuss new techniques….
5. Concise the conclusion with recent references.
Author Response
Dear Reviewer 3,
please find attached the answer
Best regards
Authors

Reviewer 4 Report
Dear Authors,
unfortunately, in my opinion, your proposal does not rise at the level required for a journal. I could not identify any piece of novelty. By the way, you, yourself do not claim any novelty.
Almost everything you preasent are nothing but ascertainments. What you claim to be a case study is not supported by concret data. A case study is based on a real fact, what does not seem to be iin your case.
Conclusion does not reveal any novelty as an achievement of your work.
Some more specific remarks on your text can be found in the attached file.
Author Response
Dear Reviewer 4,
please find attached answer
Best regards
Authors

Round 2
Reviewer 4 Report
Dear Authors,
despite you declaring that the suggestions/requests of the reviewer were treated, that did not happen. For instance, the statements referred in the reviewer's recommendations are displayed in the same shape as in the previous version of the manuscript: Many academic publications refer detect–repair as a …. Please specify it / cite sources. (line 103), will be created in the future suggestion: replace will be created with will occur. Creating is something intended. (line 111), Table 1: what is the unit measure for Avg lead time? Secs, mins? …
You declare that this is a case study. Yet, the conclusions do not present any measurable output of the comparison you claim.
So, I keep my first decision
Author Response
Dear Reviewer 4,
please find attached answers to your reviews.
Best regards
Authors
